# Ocular Surface Changes Associated with Ophthalmic Surgery

**DOI:** 10.3390/jcm10081642

**Published:** 2021-04-12

**Authors:** Lina Mikalauskiene, Andrzej Grzybowski, Reda Zemaitiene

**Affiliations:** 1Department of Ophthalmology, Medical Academy, Lithuanian University of Health Sciences, 44037 Kaunas, Lithuania; lpozeraite@gmail.com; 2Department of Ophthalmology, University of Warmia and Mazury, 10719 Olsztyn, Poland; ae.grzybowski@gmail.com; 3Institute for Research in Ophthalmology, Foundation for Ophthalmology Development, 61553 Poznan, Poland

**Keywords:** dry eye disease, ocular surface dysfunction, cataract surgery, phacoemulsification, refractive surgery, trabeculectomy, vitrectomy

## Abstract

Dry eye disease causes ocular discomfort and visual disturbances. Older adults are at a higher risk of developing dry eye disease as well as needing for ophthalmic surgery. Anterior segment surgery may induce or worsen existing dry eye symptoms usually for a short-term period. Despite good visual outcomes, ocular surface dysfunction can significantly affect quality of life and, therefore, lower a patient’s satisfaction with ophthalmic surgery. Preoperative dry eye disease, factors during surgery and postoperative treatment may all contribute to ocular surface dysfunction and its severity. We reviewed relevant articles from 2010 through to 2021 using keywords “cataract surgery”, ”phacoemulsification”, ”refractive surgery”, ”trabeculectomy”, ”vitrectomy” in combination with ”ocular surface dysfunction”, “dry eye disease”, and analyzed studies on dry eye disease pathophysiology and the impact of anterior segment surgery on the ocular surface.

## 1. Introduction

Dry eye disease (DED) is a common condition, which usually causes discomfort, but it can also be an origin of ocular pain and visual disturbances. Ocular surface inflammation is thought to be the main factor in the pathogenesis of DED. It has many overlapping causes, such as ocular surgery, environmental triggers, medication use and systemic diseases. Ophthalmic surgery may induce or worsen existing DED symptoms usually for a short-term period [1]. Ocular surface abnormalities have been reported to develop after surgical interventions, such as phacoemulsification. Despite good visual gain, DED symptoms can significantly affect quality of life and, therefore, lower a patient’s satisfaction with ophthalmic surgery [2]. The transection of corneal nerves during incision, preservatives of medications used during and after surgery, intraoperative irrigation of ocular surface and light exposure from a microscope may all contribute to ocular surface dysfunction. Although DED is mostly described after refractive and cataract surgery, other interventions, such as trabeculectomy and vitrectomy, may also alter ocular surface parameters. The prevalence of DED has increased in recent years mostly due to different environmental factors. Moreover, age is a known risk factor for DED, and elderly patients often need ocular surgery. In this review, we aim to analyze the pathogenesis of DED and ocular surface alterations of patients who underwent ophthalmic surgeries: cataract surgery, refractive surgery, trabeculectomy and vitrectomy.

## 2. Materials and Methods

PubMed, BMJ Journals, Clinical Key and the Cochrane Library were searched for entries from 2010 through to 2021 for studies which evaluated ocular surface alterations and DED symptoms after anterior segment surgery, such as cataract surgery, refractive surgery, trabeculectomy and vitrectomy. The search was filtered to include only articles written in English, and nonrelevant studies were excluded. Search terms included “cataract surgery”, “phacoemulsification”, “refractive surgery”, “trabeculectomy” and “vitrectomy” in combination with “ocular surface dysfunction” and “dry eye disease”. The search of the databases resulted in 578 entries. Duplicates (*n* = 274) were removed using Endnote software. In total, 304 records of titles and abstracts were screened, and 79 studies were included in this review. Studies were excluded when they were not relevant to the topic or focused on the various treatments or management of DED (Figure 1).

## 3. Dry Eye Disease

Tear Film and Ocular Surface Society (TFOS) Dry Eye WorkShop (DEWS) II defines DED as follows “Dry eye is a multifactorial disease of the ocular surface characterized by a loss of homeostasis of the tear film, and accompanied by ocular symptoms, in which tear film instability and hyperosmolarity, ocular surface inflammation and damage, and neurosensory abnormalities play etiological roles” [3]. DED can be classified into two categories: aqueous-deficient and evaporative. Aqueous-deficient DED may be associated with lacrimal gland dysfunction, while the evaporative type occurs with conditions affecting the eyelids or ocular surface. Eye structures and accessory organs of the eye, such as cornea, conjunctiva, eyelids, eyelashes, lacrimal and Meibomian glands, define the ocular surface. The tear film is a very important structure, as it is the interface of the ocular surface and the environment. Its function is to protect, lubricate the ocular exterior and form a stable refracting surface [4]. Chen et al. measured precorneal tear film thickness to be approximately 1.9 ± 0.9 μm using ultrahigh resolution optical coherent tomography [5]. Tear film is thought to comprise three layers: mucin, aqueous and lipid. Transmembrane mucins which are attached to epithelial cells extend to aqueous components containing various peptides, and proteins protect epithelial cells [6]. The tear film lipid layer is outermost, approximately 42 nm thick and ensures resistance to evaporation [7].

Disturbance of the balance of the tear film may be among the main factors in the development of the DED (Figure 2). Desiccation of the ocular surface, changes in tear composition and osmolarity lead to the activated production of inflammatory mediators and inflammatory cell recruitment. These processes trigger the dissolution of cell junctions, death of epithelial cells and destabilization of the tear film, which creates a cycle of inflammation [8]. Ocular surface inflammation alters the sensitization of nociceptors and causes a sensation of pain and dryness of the eye. Long-term inflammation and nerve injury can cause neuropathic pain on the ocular surface [9].

Homeostasis of the ocular surface is controlled by the physiological system and corneal innervation. The cornea is the most densely innervated organ in the human body. The ocular surface is supplied by sensory neurons of the trigeminal nerve (cranial nerve V). Sensory fibers travel to the cornea via the nasociliary branch of the ophthalmic nerve and via long ciliary nerves [9]. Long and short ciliary nerves form the pericorneal plexus at the corneal limbus. Corneal nerve fibers distribute into stromal and subepithelial plexuses below Bowman’s layer after entering the cornea in the corneoscleral limbus. Marfurt et al. analyzed the corneal nerve architecture of donor corneas. Nerve bundles entering the cornea through the corneoscleral limbus distribute uniformly around the cornea with a mean distance of 0.48 ± 0.40 mm [10]. Fibers end in the corneal epithelium as unmyelinated nerve endings which serve as nociceptors. Nociceptors are sensory to mechanical, chemical and thermal stimulation and transmit information to the brain. Facial nerves contain motor and autonomic fibers which control blinking and stimulate tear secretion in the lacrimal gland. Sensory nerve endings are able to respond to environmental changes and alter the intensity of tear production by trigeminal–parasympathetic reflex. Neural response mechanisms also regulate the secretion of compounds produced by goblet cells and Meibomian glands [11]. Due to reduced tear secretion, the tear film thins, and the corneal surface becomes vulnerable and exposed to unfavorable conditions. Corneal epithelium and terminal nerve branches may become injured. This initiates a cycle of nerve alternation which leads to the abnormal structure of corneal and conjunctival nerve fibers [9]. In a comparative study conducted by Labbe et al., the ocular surfaces of patients with non-Sjogren DED were analyzed with in vivo confocal microscopy and had a lower density and increased tortuosity and number of beadings of subbasal corneal nerves compared to control subjects. Corneal nerve density correlated with age, the Oxford scale and central corneal sensitivity. Alterations of subbasal corneal nerves were shown to be related to the severity of DED [12]. In another study, corneal sensitivity was significantly decreased in patients with DED and correlated positively with the density and number of subbasal nerves [13]. Subbasal nerve fiber length (SNFL) was also found to be significantly lower in patients with DED compared with controls. Patients with almost normal SNFL were shown to respond better to DED treatment (artificial tears and loteprednol etabonate with or without tobramycin 0.3% daily for 4 weeks) than subjects with low baseline SNFL [14]. As SNFL might help in evaluating the response to treatment, corneal subbasal nerve density, on the other hand, was found to correlate with endothelial cell loss. Endothelial cell density was lower in subjects with DED, and it correlated with the severity of the disease [15]. In a study conducted by Kheirkhan et al., patients with DED had lower densities of corneal endothelial cells and subbasal nerves compared to controls at their initial visit. The mean follow-up time was 33.2 months (±10.2). At the last visit, corneal subbasal nerve density did not change significantly, but there was a decrease in endothelial cell density, which correlated negatively with initial subbasal nerve density. Therefore, patients with low subbasal nerve density were at higher risk for endothelial cell loss [16].

In a study conducted by Kheirkhan et al., patients with DED had a higher density and size of dendritic cells and a higher number of dendrites compared to controls. These changes in the density and morphology of corneal epithelial immune dendritic cells reflected inflammation in DED [17]. The inflammatory component in the etiology of contact lens-induced DED was suggested by the increased mean Langerhans cell density in the central cornea and nasal bulbar conjunctiva found in symptomatic contact lens-wearing patients [18]. Inflammation is thought to be very important in the pathogenesis of DED. The release of mediators such as eicosanoids, bradykinin, histamine, purines and cytokines–interleukins contributes to the activation and sensitization of nociceptor terminals. Persistent exposure to inflammatory components may damage corneal nerves, while acute nerve injury causes local inflammation. These overlapping mechanisms increase the activity of sensory neurons and trigger a pain sensation, which can persist for prolonged periods of time. Twelve inflammatory cytokines (IL-1β, IL-6, IFN-γ and TNF-α) are most frequently detected in the tears of dry eye patients and are reliably used as biomarkers of DED [19]. The conjunctival expression of matrix metalloproteinase 9 and transglutaminase 2 was shown to be higher in patients with Meibomian gland dysfunction and showed inflammation of epithelial damage due to impaired tear secretion [20]. Patients with Meibomian gland dysfunction-related DED also had higher levels of inflammatory molecules, such as IL-6, IL-8, IFN-γ and ICAM-1, in their tears compared to control subjects [21]. In a study conducted by Macri et al., patients with DED were also shown to have increased levels of oxidative stress by assessing lipid peroxidation with the LP-CHOLOX test [22]. In another study, the expression of lipid peroxidation markers, such as hexanoyl-lysine, 4-hydroxy-2-nonenal and malondialdehyde, were measured in tears using enzyme-linked immunosorbent assays and, in the conjunctiva, using imunohistochemistry. In DED patients, concentrations of these markers were elevated and correlated with the severity of the disease [23].

## 4. Dry Eye Disease Diagnostic Tests

Based on the TFOS DEWS II Diagnostic methodology report, the diagnosis of DED is made when there are characteristic symptoms and at least one changed marker of ocular surface homeostasis. Ocular surface dysfunction is a prevalent condition and is associated with ocular symptoms, such as foreign body sensation, dryness, fatigue and red or watery eyes. DED symptoms can significantly affect quality of life. Symptoms are captured using standardized questionnaires, such as the ocular surface disease index (OSDI) and dry eye questionnaire (DEQ-5). The OSDI questionnaire includes 12 questions related to visual disturbance or visual function and vision-related quality of life, while DEQ-5 includes five questions about visual disturbance and its frequency. A positive result is an OSDI score ≥ 13 and DEQ-5 score ≥ 5. DED symptoms and at least one positive homeostasis test result (tear break-up time (TBUT), osmolarity or corneal staining) would constitute the diagnosis of DED. However, it should be noted that some patients may experience symptoms with no clinical signs. A positive finding of TBUT was reported to be a value < 10 s. Fluorescein instillation is usually used measuring tear film breakup time over the whole surface. Another homeostasis test is tear osmolarity. A positive test result is considered to be ≥308 mOsm/l in either eye or if the difference between two eyes is >8 mOsm/l. Ocular surface staining is assessed using fluorescein dye, 1 or 3 min after instilllation. More than five corneal spots (or >9 conjunctival spots) is considered a positive result. DED according to the OSDI score and ocular discomfort symptoms can be divided into three categories: mild (13–22), moderate (23–32) and severe (>33) DED. Osmolarity may also help to differentiate between these categories of DED, as it increases with the severity of the disease (normal: 302.2 ± 8.3 mOsm/L; mild to moderate: 315.0 ± 11.4 mOsm/L; and severe: 336.4 ± 22.3 mOsm/L) [24]. Japanese diagnostic criteria for DED (2006) consisted of three components: subjective symptoms, tear functions (either Schirmer’s test I ≤ 5 mm (5 min), TBUT ≤ 5 s) and vital staining (lisamine green, rose bengal or fluorescein staining in more than 3 points out of 9) [25]. In 2016, a new DED definition was proposed. According to the new criteria, DED symptoms and decreased TBUT (≤5 s) determine the diagnosis. Schirmer’s test value and epithelial damage are not important, as an unstable tear film is considered a central component of dry eye pathogenesis [26]. Several authors have shown that objective signs and symptoms of DED do not correlate well. Begley et al. reported low to moderate correlations between DED symptoms and results of objective tests, such as Schirmer’s test and TBUT [27]. Gupta et al. showed that 83% of patients who had no previous ocular surface disease and no symptoms before cataract surgery had abnormal osmolarity or MMP-9 test [28]. In the case of patients with chronic symptoms without or with limited clinical signs of DED, which are resistant to treatment, and neuropathic pain should be considered. Asymptomatic patients with clinical signs unreferrable to other ocular surface diseases might be offered treatment. Ocular surface condition differentiation can be complicated, as some ocular surface diseases can be comorbid with DED.

## 5. Influence of Dry Eye Disease on Quality of Vision

Healthy and stable tear film is important for good optical quality, as tear film is the first surface for light to pass into the eye. Visual acuity is often normal, but other tests may show abnormalities, such as increased optical aberrations and decreased corneal sensitivity, in patients with DED [29]. The objective scatter index (OSI) is used to quantify ocular transparency. The scatter of light moving in the direction of the retina can be caused by unstable tear film. Su et al. analyzed the tear film objective scatter index (TF-OSI) in DED. The mean value and dispersion of TF-OSI were higher in patients with DED than in healthy subjects [30]. Another study conducted by Ma et al. examined visual quality in DED. The OSI was seen to correlate with clinical symptoms and signs, such as OSDI and TBUT. OSI was also found to correlate with corneal nerve length, which suggests that nerve changes may influence poor visual quality in DED patients [31]. Herbaut et al. also found that OSI correlated with OSDI, TBUT and Schirmer’s test results in DED patients [32]. Gao et al. examined optical quality in patients before and after dry eye treatment (hyaluronate and fluorometholone eye drops for two weeks). The OSI, contrast sensitivity, corneal high order aberrations, standard deviation of corneal power and surface asymmetry index improved after the treatment [33]. Similar results were seen in a study conducted by Lu et al. Optical quality was seen to improve by evaluating high-order aberrations of corneal surface before and after DED treatment (artificial eye drops for two weeks) [34]. On the other hand, Arikan et al. compared contrast sensitivity and corneal aberrations in patients with primary Sjogren syndrome (DED) and healthy participants. Contrast sensitivity function and corneal high-order aberrations did not differ between patients with Sjogren syndrome and the control group [35]. Ocular surface dysfunction might also affect preoperative cataract surgery planning due to topography and keratometry measures [36]. Increased osmolarity of the tear film as a result of ocular surface disease has been shown to influence keratometry results and, therefore, intraocular lens (IOL) power calculations. Hyperosmolarity is associated with tear film instability and rapid breakup after blinking. Epitropolous et al. compared the keratometry value, corneal astigmatism and IOL power between hyperosmolar (osmolarity more than 316 mOsm/Lin at least one eye) and normal (osmolarity less than 308 mOsm/L in both eyes) patients. A higher variability in the keratometry value and a greater difference in the measured corneal astigmatism and IOL power were seen in the hyperosmolar group [37]. Variability of keratometry values in dry eyes was also shown in a study conducted by Roggla et al. [38]. The instillation of artificial tears before keratometry may show more stable results, although in other studies, eye drops did not the change variability of keratometry values [39].

## 6. Dry Eye Disease after Ocular Surgery

Ophthalmic surgery may cause or worsen existing DED symptoms. Corneal nerve damage during incision, perioperative medications, intraoperative irrigation of ocular surface and light exposure from a microscope and inflammatory mediators may all contribute to ocular surface dysfunction.

### 6.1. Cataract Surgery

Conventional phacoemulsification and intraocular lens (IOL) implantation are among the most common procedures of anterior segment surgery performed with over 20 million procedures performed each year globally [40]. Some patients might subsequently develop DED postoperatively. Studies show that patient satisfaction with cataract surgery is closely associated with DED symptoms, rather than objective measures of postoperative visual acuity and signs of DED [41]. The impact of the cataract surgery on the ocular surface is presented in Table 1.

Kohli et al. evaluated DED symptoms and signs after cataract surgery. This study included patients with no DED before surgery. Ocular surface dysfunction symptoms were evaluated using the OSDI questionnaire; 2 weeks after surgery, 32% of patients had an OSDI score greater than 33. Dry eye signs, such as Schirmer’s test results and TBUT, were all abnormal in 48% of patients at two weeks postsurgery. In a period of six weeks, recovering trends of DED symptoms were seen [42]. In another study conducted by Sahu et al., TBUT and Schirmer’s test values significantly decreased 5 days postoperatively, although a rising trend was seen approximately two months after surgery [48]. Kasetsuwan et al. reported the incidence of DED (OSDI score more than 25) after cataract surgery to be 9.8% one week postoperatively. No correlation was found between DED postoperatively and the sex or age of the patients [1]. Mild ocular discomfort may occur before cataract surgery, but DED is usually not diagnosed. This is associated with an incomplete preoperative evaluation of the ocular surface due to a lack of time or other reasons. Although patients may not make complaints related to dry eyes, results of the tear osmolarity, TBUT and Schirmer’s test may be abnormal. Trattler et al. evaluated the incidence of DED in patients before cataract surgery. Although 60% did not feel ocular discomfort, such as foreign body sensation, almost 63% had a TBUT ≤ 5 s, and 77% of eyes had corneal staining [49]. Cho and Kim studied changes in the ocular surface after cataract surgery in patients with and without DED preoperatively. In the non-DED group, TBUT and Schirmer’s test results were significantly worse postoperatively; in the DED group, changes in dry eye tests were not significantly different. There was also no difference in dry eye test results according to incision location [50]. Oh et al. demonstrated a corneal sensitivity decrease one day postoperatively, which returned to almost preoperative levels, although Schirmer’s test results were not very different from preoperatively [43]. Although many studies show short-term ocular surface dysfunction, persistent pain may occur. Persistent postsurgical pain, which is described as pain that continues for 3 to 6 months following surgery, can be associated with surgical nerve injury [51]. In a study conducted by Xue et al., persisting DED symptoms were reported for more than 3 months. In this study, TBUT and Schirmer’s test values were seen to decrease at 1 month postoperatively and gradually increase up to 6 months postoperatively. DED symptoms, such as burning and hypersensitivity to wind and light, are similar to other neuropathic pain syndromes [52]. Sajnani et al. interviewed patients who underwent standard phacoemulsification approximately 5 to 7 months after surgery and reported a 34% frequency of persistent pain syndrome. Female sex, presence of an autoimmune disease and previously diagnosed DED were factors that increased the likelihood of developing persistent postoperative pain after cataract surgery [51]. Corneal nerve damage due to incision may reduce tear production and disturb normal evaporation, which can lead to inflammation of the ocular surface. Multiple inflammatory cytokines and chemokines cause cell impairment and sets a cycle of damage and inflammation. Goblet cell density and differentiation might be affected, as shown in a study by Kohli et al. [42].

Light exposure and ultrasound energy used during phacoemulsification could affect the ocular surface through photochemical damage. The production of reactive oxygen compound free radicals causes damage to the corneal and conjunctival epithelial cells. Sahu et al. showed a negative correlation of microscopic light exposure and cumulative dissipated energy with TBUT and Schirmer’s test results, and a positive correlation with DEQ-5 score, although none of these associations were statistically significant. For all dry eye signs which were tested—Schirmer’s test and TBUT—the tear meniscus height showed deterioration 5 days postoperatively compared to the preoperative values followed up a gradual increase up to two months after surgery [48].

Corneal astigmatism in patients undergoing cataract surgery may be treated by placing corneal incisions near the limbus–limbal relaxing incisions (LRIs) or by the implantation of toric intraocular lens. LRIs with simultaneous phacoemulsification could worsen postoperative ocular surface changes. Incisional corneal procedures may damage nerve fibers, which leads to decreased tear production, corneal sensitivity and ocular discomfort. Ahmed et al. evaluated TBUT and Schirmer’s test results postoperatively between two groups of patients who had vertical or horizontal LRIs. Schirmer’s test results did not change significantly after the surgery in both groups (except in the first postoperative week in the horizontal LRI group), although TBUT was significantly reduced. The orientation of LRI did not significantly impact these changes [53].

Femtosecond laser-assisted cataract surgery (FLACS) could also cause or worsen DED symptoms and signs. Ju et al. analyzed the tear film stability, Schirmer’s test and OSDI score before FLACS and 1 day, 1 week, 1 month and 3 months after the procedure. All parameters were worse after the surgery, although tear film break-up time returned to basic levels 1 month postoperatively and Schirmer’s test at 3 months postoperatively. For DED symptoms, the OSDI score was higher after the procedure and did not return to the preoperative level in 3 months [45]. Yu et al. compared the OSDI, Schirmer’s test and fluorescein staining of the cornea in patients after FLACS and conventional phacoemulsification after 1 day, 1 week and 1 month. Both surgeries worsened ocular surface dysfunction, although FLACS had a higher risk of DED symptoms and corneal staining [46]. Similar results were seen in a study performed by Shao et al.—OSDI score, tear film break-up time and Schirmer’s test results were worse after the FLACS and phacoemulsification, but FLACS’s effect on the ocular surface was greater than that of phacoemulsifiaction [54]. In another study conducted by Schargus et al., tear film parameters, such as tear film osmolarity, Schirmer’s test and MMP-9 concentration, were compared 1 and 3 months after patients underwent FLACS or phacoemulsification. No significant difference in tear film parameters was observed between the two techniques [47].

Cataract surgery usually causes short-term alterations, although patients with preoperative ocular surface dysfunction are more likely to develop DED for a longer time [44]. FLACS can cause a greater impact on the ocular surface compared to conventional phacoemulsification.

### 6.2. Refractive Eye Surgery

Another type of anterior segment surgery that might induce ocular surface dysfunction is refractive eye surgery [55]. (Table 2) Corneal refractive surgery may cause neuropathic DED, the incidence of which decreases in the first six months postoperatively [56]. In a systematic review and meta-analysis conducted by Sambhi et al., DED rates were compared between different refractive surgeries. A significant reduction in tear production and TBUT after the procedure was shown with laser in situ keratomileusis (LASIK). No significant reduction was shown after femtosecond lenticule extraction (FLEX), small incision lenticule extraction (SMILE) and photorefractive keratectomy (PRK) [57]. Postoperative DED after LASIK is thought to occur most commonly out of all ophthalmic surgeries. Although symptoms usually last a month postoperatively, few patients experience discomfort for more than a year after the procedure [58]. Post-LASIK DED is thought to be a neuropathic-related disease [59]. Other refractive surgery can also cause corneal nerve damage and neuropathic corneal pain [60]. Vestergaard et al. compared corneal subbasal nerve morphology, corneal sensation, tear osmolarity, tear film break-up time and Schirmer’s test among patients who underwent FLEX and small SMILE. Six months after surgery, the total number of nerves and subbasal nerve density were decreased in both groups, although changes were more significant in the FLEX group. Corneal sensation was lower in the FLEX group; tear film evaluation results were not significantly different in groups after 6 months [61]. Some authors compare femtosecond-assisted LASIK (FS-LASIK) and SMILE impacts on the ocular surface. Recchioni analyzed short-term (1 month postoperatively) clinical outcomes between these two procedures and found that both provided good visual results, although FS-LASIK increased DED symptoms more. Other ocular surface characteristics, such as tear meniscus height, tear osmolarity and noninvasive breakup time, did not differ significantly in a 1-month period. Corneal nerve density was affected more in the FS-LASIK group [62]. Kobashi et al. compared ocular surface parameters in patients who underwent SMILE and LASIK. OSDI scores, tear film break-up time, corneal sensitivity and corneal subbasal nerve density at 6 months postoperatively were significantly worse in the LASIK group [63]. Similar results were seen in a study conducted by Denoyer et al.—in a 6-month period, SMILE had a lower impact on the ocular surface with regard to DED incidence, corneal sensitivity and corneal nerve density than LASIK [64]. In meta-analysis conducted by He et al., corneal sensitivity was shown to be worse in the LASIK group compared to SMILE at 1 week, 1 month and 3 months after surgery; however, after 6 months, sensitivity was similar in both groups [65]. Wang et al. also compared DED symptoms and signs following SMILE and LASIK and found that SMILE produced less DED than LASIK at 6 months postoperatively, although after 12 months, results were not very different. [66] However, in a study performed by Hassan et al., no tear film tests were abnormal before and after LASIK (1, 30 and 60 days after surgery) [67].

A retrospective study, conducted by Shehadeh-Mashor, showed that factors associated with clinically significant DED after keratorefractive surgery (LASIK or photorefractive keratectomy) were older age, female gender and lower preoperative refractive error [68]. Li et al. evaluated tear film function in refractive surgical candidates and found that a high proportion of them had preexisting DED. Furthermore, 41% were diagnosed with DED with the most common abnormal finding of tear film instability, and 45% of refractive surgical candidates had a history of contact lens wearing [69]. As with cataract surgery, patients who have preoperative DED may be subjected to treatment with artificial tears before refractive surgery. Appropriately selected patients usually achieve good visual outcomes. According to a study by Torricelli et al., 3.7% of patients who had contraindications for refractive surgery (LASIK or PRK) had severe DED, which did not respond to treatment [70].

### 6.3. Trabeculectomy

Trabeculectomy is among the most common surgeries for glaucoma treatment. It has been observed that this surgery may affect the surface of the eye, but limited research has been conducted to describe this (Table 3). Most authors suggest that changes in the ocular surface may be caused due to the application of antimetabolic drugs and avascular postoperative filtering blebs [71]. On the other hand, glaucoma surgery may bring long-term benefits to the ocular surface, as there is no further need for intraocular pressure-lowering medications anymore [72]. The long-term use of antiglaucomatous eye drops may cause ocular discomfort symptoms, Meibomian gland loss and tear film dysfunction. Trabeculectomy may also become a cause of Meibomian gland loss. Sagara evaluated Meibomian gland morphology and loss 7.4 years on average after trabeculectomy. Patients included in the study had a history of antiglaucomatous drop use containing Benzalkonium chloride for 0.1 to 32.5 years. No significant correlation between meiboscores and the duration of topical medication usage was found. Blebs with mitomycin C, particularly when the bleb is avascular, may cause Meibomian gland loss and lead to ocular surface damage [73]. Zhong et al. analyzed the influence of trabeculectomy on the ocular surface at 3 days, 1 month and 3 months postoperatively. Patients using topical medications with Benzalkonium were excluded. Ocular surface parameters, such as noninvasine keratography tear film break-up time, at 3 months postoperatively did not recover to preoperative levels. Tear meniscus height returned to baseline at 1 and 3 months postoperatively [74]. In a study conducted by Vaajanen et al., patients with a previous history of topical antiglaucomatous drug use for 8.1 ± 6.8 years underwent successful trabeculectomy. The termination of topical antiglaucoma medication was beneficial to the ocular surface. Although conjunctival redness and irritation were reduced during a follow-up period of 1 year after trabeculectomy, tear production did not change [75].

### 6.4. Vitrectomy

Vitrectomy-associated ocular surface disturbances and changes are not well known. The size and structure of the incision, as well as the surgery technique, are thought to have an effect on ocular surface changes (Table 3). Ghasemi et al. evaluated Schirmer’s test I results 1 and 3 months after 20- and 23-gauge vitrectomy. Schirmer’s test values were significantly decreased in both groups 1 and 3 months after surgery, although 3 months after surgery, Schirmer’s test measurements were significantly lower in the 20-gauge vitrectomy group [76]. In another study conducted by Nemcansky et al., tear film osmolarity was measured 10 and 30 days after 25-gauge pars plana vitrectomy. No significant change was found in tear osmolarity postoperatively. Sato et al. investigated ocular surface changes before and 1 week and 1 month after 25-gauge vitrectomy (various indications). Combined cataract surgery was performed in 71% of cases. TBUT and corneal fluorescein staining score did not change significantly, although conjunctival fluorescein staining score and tear meniscus height were higher 1 week postoperatively but returned to baseline after 1 month. The suturing of sclerotomy also contributed to increased tear meniscus height postoperatively. It is thought that absorbable sutures (8-0 Vicryl suture) may cause irritation of the ocular surface [78]. Lee et al. analyzed the effects of the suturing of sclerotomies during 23-gauge pars plana vitrectomies on the ocular surface 1 week, 1 month and 3 months postoperatively. The study showed that suturing had a negative effect on TBUT 3 months postoperatively and also caused ocular surface discomfort, as shown by a higher OSDI score [77]. Mani et al. evaluated DED symptoms and signs in patients 8 weeks after vitreo-retinal surgeries (scleral buckle and microincision vitrectomy surgery). Although the clinical symptoms of DED did not change significantly, there was a lower conjunctival goblet cell density, increased tear cytokines (IL-1RA, IL-5, IL-9, FGF, PDGFbb, IL2, IL6, IL15, GMCSF and IFNg) and altered gene expression of mucins and aquaporins. These studies demonstrated that ocular surface changes could be seen at the molecular level [79].

## 7. Take Home Messages

DED is a multifactorial disease of the ocular surface with etiology of tear film instability and hyperosmolarity, ocular surface inflammation, damage and neurosensory abnormalities, which often presents with ocular symptoms.DED is diagnosed when there are characteristic symptoms, confirmed by standardized questionnaires (OSDI score ≥ 13 or DEQ-5 score ≥ 5) and at least one marker of ocular surface homeostasis disruption (TBUT < 10 s; tear osmolarity ≥ 308 mOsm/L in either eye or if the difference between two eyes is >8 mOsm/L; ocular surface staining of more than five corneal spots (or >9 conjunctival spots)).The ocular surface should be carefully evaluated before surgery to predict the exacerbation of DED and to optimize the ocular surface for a better visual outcome. For patients with preoperative dry eye, ocular surface management before surgery should be considered (tear supplements and ointments; anti-inflammatory agents).Cataract surgery usually causes short-term ocular surface changes. Femtosecond laser-assisted cataract surgery can cause a greater impact on the ocular surface compared to conventional phacoemulsification.DED is thought to occur most commonly after the LASIK procedure. Nonsignificant changes in postoperative tear production were seen in other refractive surgery types, such as SMILE, FLEx and PRK.Trabeculectomy and vitrectomy seem to alter tear film stability and cause changes to ocular surface parameters; however, few studies have been conducted to describe it.

## Figures and Tables

**Figure 1 jcm-10-01642-f001:**
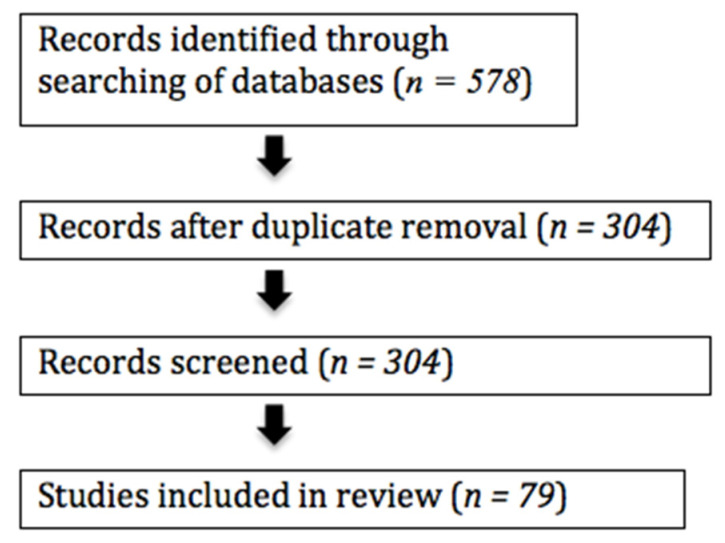
Flowchart of study selection process.

**Figure 2 jcm-10-01642-f002:**
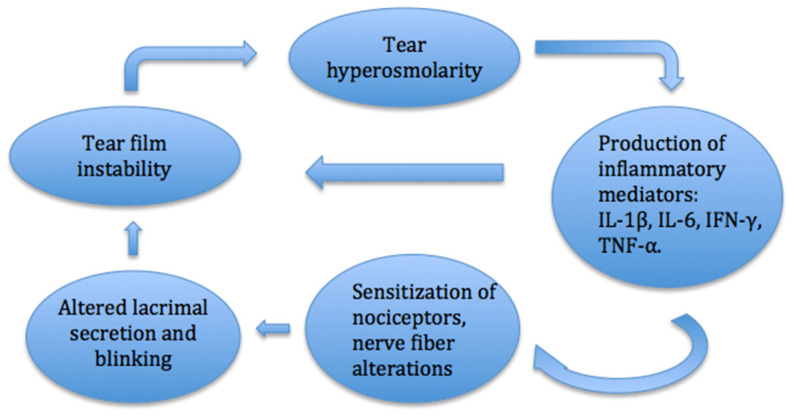
Pathophysiology of dry eye disease.

**Table 1 jcm-10-01642-t001:** Ocular surface dysfunction after cataract surgery.

Study	Design	Surgical Procedure	Postoperative Tests	Results
Kohli et al., 2019 [42]	Prospective study	Phacoemulsification	2, 4 and 6 weeks postoperatively	OSDI score ≥ 33 was 32% at postoperative 2 weeks and 14.0% at 6 weeks. Patients with abnormal dry eye signs, such as SI-T score ≤ 10 and TBUT ≤ 10 at 2 weeks, were 48.0% and 48.0%, and at 6 weeks postoperatively—24.0% and 28.0%, respectively.
Oh et al., 2012 [43]	Randomized controlled trial	Phacoemulsification	1 day, 1 month, and 3 months after surgery	Corneal sensitivity at the center and temporal incision sites, TBUT decreased significantly at 1 day postoperatively but returned to almost the preoperative level 1 month postoperatively. Mean goblet cell density (GCD) decreased significantly at 1 day, 1 month and 3 months postoperatively.
Zamora et al., 2020 [44]	Prospective interventional study	Phacoemulsification	1 day, 1 week, and 1 month postoperatively	OSDI score increased significantly 1 week postoperatively. TBUT and Schirmer’s test values were decreased 1 month after surgery.
Ju et al., 2019 [45]	Prospective study	FLACS	1, 7, 30 and 90 days after surgery	OSDI scores after surgery were higher but did not return to the basic level by 3 months. TBUT decreased 1 week postoperatively, but returned to basic levels at 1 month; Schirmer’s test results returned to preoperative levels at 3 months.
Yu et al., 2015 [46]	Prospective consecutive nonrandomized comparative cohort study.	FLACS or phacoemulsification	1 day, 1 week, and 1 month postoperatively	OSDI and fluorescein staining scores elevated from baseline; TBUT and Schirmer’s test I values decreased in 1 week, but did not return to basic levels at 1 month postoperatively. OSDI score was greater in the FLACS group at 1 week.
Schargus et al., 2020 [47]	Prospective, randomized, single-center study	FLACS or phacoemulsification	1 and 3 months postoperatively	No statistically significant difference was found in regard to tear film osmolarity, Schirmer’s test and MMP-9 concentration between the two groups.

FLACS: Femtosecond laser-assisted cataract surgery; OSDI: ocular surface disease index; TBUT: tear break-up time.

**Table 2 jcm-10-01642-t002:** Ocular surface dysfunction after refractive eye surgery.

Study	Design	Surgical Procedure	Postoperative Tests	Results
González-García et al., 2020 [56]	Prospective, longitudinal study	Surface ablation refractive surgery	1, 3 and 6 months postoperatively	IL-4, IL-5, IL-6, IL-13, IL-17A and IFN-γ tear levels were significantly increased 1, 3 and 6 months postoperatively.
Chao et al., 2015 [59]	Prospective longitudinal cohort study	LASIK	1 day, 1 week, 1 month, 3 months and 6 months postoperatively	DED symptoms (ocular comfort index) and noninvasive TBUT did not change postoperatively. Central corneal sensitivity did not return to preoperative levels by 6 months. Corneal nerve morphology (nerve fiber density, number of interconnections and average nerve fiber width) decreased immediately, and did not return to preoperative levels by 6 months post-LASIK.
Vestergaard et al., 2013 [61]	Randomized controlled trial	FLEX in one eye and SMILE in the other	6 months after surgery.	No difference was found in tear osmolarity, noninvasive TBUT (keratograph), tear meniscus height (anterior segment OCT), Schirmer’s test and fluorescein TBUT between FLEX and SMILE. Corneal subbasal nerve density and total nerve number decreased 6 months after surgery, although total number of nerves decreased more in FLEX eyes than in SMILE eyes.
Hassan et al., 2013 [67]	Comparative study	LASIK	1, 30 and 60 days after the surgery.	No significant change was observed in the values of Schirmer’s test, corneal staining and TBUT during the follow-up period.
Wang et al., 2015 [66]	Prospective, nonrandomised, observational study	SMILE, FS-LASIK	1, 3, 6 and 12 months postoperatively.	TBUT reduced 1 and 3 months after SMILE, 1, 3 and 6 months following FS-LASIK. Dry eye symptoms were increased 1 and 3 months following SMILE, and 1, 3 and 6 months following FS-LASIK.

LASIK: laser in situ keratomileusis; FS-LASIK: femtosecond-assisted laser in situ keratomileusis; SMILE: small incision lenticule extraction; FLEX: femtosecond lenticule extraction; DED: dry eye disease.

**Table 3 jcm-10-01642-t003:** Ocular surface dysfunction after trabeculectomy and vitrectomy.

Study	Design	Surgical Procedure	Postoperative Tests	Results
Sagara et al., 2014 [73]	Cross-sectional observational case study	Trabeculectomy with mitomycin C	Duration from trabeculectomy to examination was 0.7 to 24.5 years (median 7.4)	Meibomian gland loss was higher in the bleb-contacting upper eyelid areas than bleb-noncontacting, especially when the bleb is avascular.
Zhong et al., 2019 [74]	Retrospective, case–control	Trabeculectomy	3 days, 1 month and 3 months postoperatively	Noninvasive keratography TBUT was reduced at 3 months postoperatively.
Ghasemi et al., 2017 [76]	Prospective, nonrandomized, comparative study	Pars plana vitrectomy	1 and 3 months after vitrectomy	Schirmer’s test values decreased significantly at 1 and 3 months postoperatively.
Lee et al., 2019 [77]	Retrospective chart review	Pars plana vitrectomy	1 week, 1 month and 3 months after surgery.	TBUT lower 1 week postoperatively, improved 1 and 3 months after surgery. The OSDI scores worsened 1 week postoperatively and recovered at 1 and 3 months.

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
