# Peer review of "Ocular Surface Changes Associated with Ophthalmic Surgery"

_jcm, 2021, doi:10.3390/jcm10081642_

Round 1
Reviewer 1 Report
The authors reported the impact of anterior segment surgery such as cataract, refractive surgery, trabeculectomy, and vitrectomy on dry eye syndrome through an extensive review of relevant articles. I think this article provides clinically useful information to the ophthalmologist. But this reviewer has several comments to this review article.
- I think that vitrectomy is not an anterior segment surgery.
- Please provide the schematic images, which show how many article you found initially, and how many articles was excluded and why they was excluded. Such schematic image should be provided as a figure for this type review article.
Author Response
Thank you for the comments.

Reviewer 2 Report
Overall a nicely written paper which is helpful for the general literature.
1) There is no clear cut definition of dry eye the way this paper seems to convey. TFOS is simply one report of many. The diagram in Figure 2 is not conclusive but simply one example of assessing signs and symptoms. This figure and the text describing the definition is an oversimplification. In addition, beyond additional definitions there are also variations on definitions for mild, moderate, and severe dry eye
2. Reference 38 is described in the text on page 5. This is a bit misleading as osmolarity influences ocular surface which influences keratometry. The osmolarity value alone is just a number.
3. Cataract surgery - there should be additional discussion of impact of LRIs on dry eye and ocular surface, especially use of LRIs with femtosecond surgery. Also should be mention that many of these cataract patients have pre-existing dry eye and should be references that surgery can make pre-existing dry eye worse. Additional detail on this should be included.
4. In glaucoma surgery section there should also be discussion about the surface of these eye having been taking multiple ocular medications for many years and potential BAK and medication induced ocular surface toxicity.
Author Response
Response to the reviewers’ comments (Reviewer 2)
We are very pleased that the revised version of our paper is being considered for publication in the Journal of Clinical Medicine and grateful for the reviewer's comments. Please find below our point-by-point responses to the comments of the Reviewer 2:
1) There is no clear cut definition of dry eye the way this paper seems to convey. TFOS is simply one report of many. The diagram in Figure 2 is not conclusive but simply one example of assessing signs and symptoms. This figure and the text describing the definition is an oversimplification. In addition, beyond additional definitions there are also variations on definitions for mild, moderate, and severe dry eye
We used TFOS definition and diagnostic criteria of dry eye disease as this version was often used in various studies, articles and practice patterns (dry eye syndrome preferred practice pattern by AAO). Other diagnostic criteria for dry eye disease proposed by Japan Dry Eye society are also mentioned in the text. We added definitions for mild, moderate and severe dry eye:
"Dry eye disease according to OSDI score and ocular dyscomfort symptoms can be divided into three categories: mild (13-22), moderate (23-32) and severe (>33) dry eye disease. Osmolarity may also help to differentiate between these categories of dry eye disease, as it increases with severity of the disease (normal: 302.2 ± 8.3 mOsm/L, mild-to moderate: 315.0 ± 11.4 mOsm/L and severe: 336.4 ± 22.3 mOsm/L)."
2) Reference 38 is described in the text on page 5. This is a bit misleading as osmolarity influences ocular surface which influences keratometry. The osmolarity value alone is just a number.
We corrected the text to: "Increased osmolarity of the tear film as a result of ocular surface disease has been shown to influence keratometry results and therefore intraocular lens (IOL) power calculations. Hyperosmolarity is associated with tear film instability and rapid breakup after blinking. Epitropolous et al. compared keratometry value, corneal astigmatism and IOL power between hyperosmolar (osmolarity more than 316 mOsm/l in at least one eye) and normal (osmolarity less than 308 mOsm/l in both eyes) patients. Higher variability in keratometry value, greater difference in measured corneal astigmatism and IOL power were seen in hyperosmolar group."
3) Cataract surgery - there should be additional discussion of impact of LRIs on dry eye and ocular surface, especially use of LRIs with femtosecond surgery. Also should be mention that many of these cataract patients have pre-existing dry eye and should be references that surgery can make pre-existing dry eye worse. Additional detail on this should be included.
Added information about impact of different orientation of LRIs on tear film tests: "Limbal relaxing incisions (LRIs) with simultaneous phacoemulsification could worsen postoperative ocular surface changes. Incisional corneal procedures may damage nerve fibers, which leads to decreased tear production, corneal sensitivity and ocular discomfort. Ahmed et al. evaluated TBUT and Schirmer's test results postoperatively between two groups of patients who had vertical or horizontal LRIs. Shirmer's test results did not change significantly after the surgery in both groups (except in the first postoperative week in horizontal LRI group), although TBUT were significantly reduced. Orientation of LRI did not significantly impact these changes."
Using keywords "limbal relaxing incisions" in combination with "dry eye disease", "ocular surface dysfunction" for search in databases resulted in very small amount of records, which does not allow to develop this topic more widely.
Pre-existing dry eye and preoperative tests of the ocular surface are important for the postoperative course of dry eye disease, we added more information in the text.
- In glaucoma surgery section there should also be discussion about the surface of these eye having been taking multiple ocular medications for many years and potential BAK and medication induced ocular surface toxicity.
We added information about topical antiglaucomatous drugs used before surgery in the studies, where it was provided.
Reviewer 3 Report
Thank you for the opportunity to review this review of ocular surface changes associated with anterior segment surgery. The manuscript provides an overview of common anterior segment surgeries that may cause ocular surface changes. Some suggestions and requested changes are provided below in the hopes of improving the quality if the manuscript:
- This is just a suggestion, but It would be useful to convert this manuscript to a systematic review as it will provide more in-depth and detailed evidence of the studies available and the data out there.
- Throughout the manuscript the author changes from dry eye syndrome, ocular surface dysfunction, dry eye disease and dry eye. I would recommend keeping it consistent throughout the manuscript.
- There are a number of grammatical errors throughout the manuscript, some examples include:
- Line 25, there should be a ‘the’ between be main ...
- Line 56, add while between dysfunction, evaporative …
- Line 63, it should be comprised not composed.
- Line 69, there should be an ‘and’ between mediators, inflammatory cell recruitment
I would suggest all authors double check the grammar.
- The OSDI is a 12-question questionnaire not 6, (Line 150) and it assesses frequency and intensity of symptoms. DEQ-5 is a 5 question-questionnaire not 4.
- I would suggest rewording the following sentence (Line 153). “If patients has dry eye disease symptoms, one homeostasis test result should be positive…” This is not always the case; patients can experience symptoms but show no signs of dry eye. Also, it should be ‘If patients have”
- The author defined TBUT in line 155 but defines it again in line 170. Additionally, they switch between TBUT and tear break up time throughout the text. The use of acronyms for a number of words are inconsistently used throughout the manuscript not just TBUT. E.g FLACS. I would keep it consistent.
- Schirmer’s test is spelt incorrectly at certain parts of the manuscript.
- Figure 2 needs to be amended. Osmolarity criteria should be 308, not 306
- There is an error with the numbering of heading titles. Both Dry eye diagnostic tests and influence of dry eye on quality of vision is 4. This means the following numbering of headings is incorrect.
- Check reference formatting in line 234 and 247.
- Remember to use past tense in the manuscript, the author switches between present tense and past tense.
- Does the author have any theories as to why cataract surgery causes dry eye, and why FLACS has a greater impact?
- There should be a more detailed discussion on antiglaucoma medication and the impact of the ocular surface. While I understand the point of the review is to focus on anterior segment surgery, anti-glaucoma medication is usually prescribed first before surgery is even considered. Therefore, the impact on the surface of the eye is possibly caused by the drug not the surgery.
- Does the author have any possibly explanations as to why anterior segment surgery (Refractive surgery, trabeculectomy, vitrectomy) affects the ocular surface?
- Why weren’t all studies identified from the search included in table 1? Additionally, it would be more useful to separate the tables based on surgery, as opposed to combining it all together.
- The references need to be reviewed. Some references do not have the journal name. Reference 1, 3, 5, 7, 11, 12, 28, 32, 47, 49.
Author Response
Response to the reviewers’ comments (Reviewer 3)
We are very pleased that the revised version of our paper is being considered for publication in the Journal of Clinical Medicine and grateful for the reviewer's comments. Please find below our point-by-point responses to the comments of the Reviewer 3:
- This is just a suggestion, but It would be useful to convert this manuscript to a systematic review as it will provide more in-depth and detailed evidence of the studies available and the data out there.
Thank you for the suggestion.
- Throughout the manuscript the author changes from dry eye syndrome, ocular surface dysfunction, dry eye disease and dry eye. I would recommend keeping it consistent throughout the manuscript.
We kept two terms in the manuscript: dry eye disease and ocular surface dysfunction.
- There are a number of grammatical errors throughout the manuscript, some examples include:
- Line 25, there should be a ‘the’ between be main ...
- Line 56, add while between dysfunction, evaporative …
- Line 63, it should be comprised not composed.
- Line 69, there should be an ‘and’ between mediators, inflammatory cell recruitment
Thank you, we edited the manuscript according to the remarks.
I would suggest all authors double check the grammar.
- The OSDI is a 12-question questionnaire not 6, (Line 150) and it assesses frequency and intensity of symptoms. DEQ-5 is a 5 question-questionnaire not 4.
We corrected these mistakes.
- I would suggest rewording the following sentence (Line 153). “If patients has dry eye disease symptoms, one homeostasis test result should be positive…” This is not always the case; patients can experience symptoms but show no signs of dry eye. Also, it should be ‘If patients have”.
We corrected these mistakes.
- The author defined TBUT in line 155 but defines it again in line 170. Additionally, they switch between TBUT and tear break up time throughout the text. The use of acronyms for a number of words are inconsistently used throughout the manuscript not just TBUT. E.g FLACS. I would keep it consistent.
We corrected these mistakes.
- Schirmer’s test is spelt incorrectly at certain parts of the manuscript.
We corrected these mistakes.
- Figure 2 needs to be amended. Osmolarity criteria should be 308, not 306
We deleted the figure.
- There is an error with the numbering of heading titles. Both Dry eye diagnostic tests and influence of dry eye on quality of vision is 4. This means the following numbering of headings is incorrect.
We corrected these mistakes.
- Check reference formatting in line 234 and 247.
- Remember to use past tense in the manuscript, the author switches between present tense and past tense.
- Does the author have any theories as to why cataract surgery causes dry eye, and why FLACS has a greater impact?
Some of the theories mentioned in the text - "Corneal nerve damage due to incision may reduce tear production and disturb normal evaporation which can lead to inflammation of ocular surface. Multiple inflammatory cytokines and chemokines cause cell impairment and sets a cycle of damage and inflammation. Goblet cell density and differentiation might be affected as it was shown in study by Kohli et al. [43] Light exposure, ultrasound energy used during phacoemulsification could affect ocular surface through photo-chemical damage. Production of reactive oxygen compounds - free radicals causes damage to corneal and conjunctival epithelial cells."
When performing FLACS, vacuum pressure to the ocular surface and laser procedure may affect tear film and increase the risk for dry eye disease.
- There should be a more detailed discussion on antiglaucoma medication and the impact of the ocular surface. While I understand the point of the review is to focus on anterior segment surgery, anti-glaucoma medication is usually prescribed first before surgery is even considered. Therefore, the impact on the surface of the eye is possibly caused by the drug not the surgery.
We added information about topical antiglaucomatous drops usage before the surgery where it was available. In this article we tried to focus only on the impact caused by the surgery.
- Does the author have any possibly explanations as to why anterior segment surgery (Refractive surgery, trabeculectomy, vitrectomy) affects the ocular surface?
Refractive surgery: one of the possible explanations, as with the FLACS, application of the suction ring and impact on perilimbic conjunctiva might be an important factor.
Trabeculectomy: Most authors suggest that changes in ocular surface may be caused due to application of anti-metabolic drugs and avascular postoperative filtering blebs.
Vitrectomy: Vitrectomy associated ocular surface disturbances and changes are not well known. It is thought that absorbable suture (8-0 Vicryl suture) used in sclerotomies suturing may cause the irritation of the ocular surface.
- Why weren’t all studies identified from the search included in table 1? Additionally, it would be more useful to separate the tables based on surgery, as opposed to combining it all together.
We separated table 1 to three parts: cataract surgery, refractive surgery, trabeculectomy and vitrectomy. Studies included in the table 1 were more focused on the impact of the surgery and seemed more appropriate than other.
- The references need to be reviewed. Some references do not have the journal name. Reference 1, 3, 5, 7, 11, 12, 28, 32, 47, 49.
We corrected these mistakes.
Round 2
Reviewer 2 Report
Authors appropriately responded to the issues from the first review.
Author Response
Thank you for your comments.
Reviewer 3 Report
Improvements have been made to the review. Some minor grammatical comments to further improve the manuscript are below:
- Consistent use of DED. The acronym is defined at line 22 but 'dry eye disease' is used throughout the text. For example in line 37, 56, 60, 166, 184, 316, 336, 359
- Line 101: In 'a' comparative study conducted 102 by Labbe et al., ocular surface of patients with non-Sjogren DED
- Meibomian is spelt with a capital M in some sections and lower case in others. I would suggest keeping it one or the other.
- I think it's important to note that the DEWS II developed the diagnostic test battery to help clinicians identify and diagnose DED. In order to diagnose a DED patient according to the DEWS II, one would hope that a patient would have a positive homeostasis result. I would suggest rewording line 157, for example, "DED symptoms and at least one positive homeostasis test results (TBUT, osmolarity or corneal staining), would constitute to the diagnosis of DED. However, it should be noted some patients may experience symptoms with no clinical signs."
- Line 361 - is it meant to be 3.7%?
- Line 417 - 5. Take home message should be '7'
Author Response
We are grateful for the reviewer's comments to improve the manuscript. Please find below our point-by-point responses to the comments of the Reviewer 3:
- Consistent use of DED. The acronym is defined at line 22 but 'dry eye disease' is used throughout the text. For example in line 37, 56, 60, 166, 184, 316, 336, 359
'Dry eye disease' were changed to DED.
- Line 101: In 'a' comparative study conducted 102 by Labbe et al., ocular surface of patients with non-Sjogren DED
We corrected the mistake.
- Meibomian is spelt with a capital M in some sections and lower case in others. I would suggest keeping it one or the other.
We kept Meibomian with capital M throughout the text.
- I think it's important to note that the DEWS II developed the diagnostic test battery to help clinicians identify and diagnose DED. In order to diagnose a DED patient according to the DEWS II, one would hope that a patient would have a positive homeostasis result. I would suggest rewording line 157, for example, "DED symptoms and at least one positive homeostasis test results (TBUT, osmolarity or corneal staining), would constitute to the diagnosis of DED. However, it should be noted some patients may experience symptoms with no clinical signs."
Thank you, we rewrote the sentences.
- Line 361 - is it meant to be 3.7%?
Yes, we corrected the mistake.
- Line 417 - 5. Take home message should be '7'.
We corrected the mistake.